# Forecasting of influenza activity and associated hospital admission burden and estimating the impact of COVID-19 pandemic on 2019/20 winter season in Hong Kong

Yiu-Chung Lau[1,2], Songwei Shan[1,2], Dong Wang[1,2], Dongxuan Chen[1,2], Zhanwei Du[1,2], Eric H. Y. Lau[1,2,3], Daihai He[4], Linwei Tian[1], Peng Wu[1,2], Benjamin J. Cowling[1,2]*, Sheikh Taslim Ali[1,2]

1 WHO Collaborating Centre for Infectious Disease Epidemiology and Control, School of Public Health, Li Ka Shing Faculty of Medicine, The University of Hong Kong, Hong Kong Special Administrative Region, China, 2 Laboratory of Data Discovery for Health Limited, Hong Kong Science and Technology Park, New Territories, Hong Kong Special Administrative Region, China, 3 Institute for Health Transformation, School of Health and Social Development, Deakin University, Burwood, Australia, 4 Department of Applied Mathematics, Hong Kong Polytechnic University, Hong Kong Special Administrative Region, China

* bcowling@hku.hk

**Data Availability Statement:** Statistical analyses were conducted using R version 4.1.1 (R Foundation for Statistical Computing, Vienna,

## Abstract

Like other tropical and subtropical regions, influenza viruses can circulate year-round in Hong Kong. However, during the COVID-19 pandemic, there was a significant decrease in influenza activity. The objective of this study was to retrospectively forecast influenza activity during the year 2020 and assess the impact of COVID-19 public health social measures (PHSMs) on influenza activity and hospital admissions in Hong Kong. Using weekly surveillance data on influenza virus activity in Hong Kong from 2010 to 2019, we developed a statistical modeling framework to forecast influenza virus activity and associated hospital admissions. We conducted short-term forecasts (1–4 weeks ahead) and medium-term forecasts (1–13 weeks ahead) for the year 2020, assuming no PHSMs were implemented against COVID-19. We estimated the reduction in transmissibility, peak magnitude, attack rates, and influenza-associated hospitalization rate resulting from these PHSMs. For short-term forecasts, mean ambient ozone concentration and school holidays were found to contribute to better prediction performance, while absolute humidity and ozone concentration improved the accuracy of medium-term forecasts. We observed a maximum reduction of 44.6% (95% CI: 38.6% - 51.9%) in transmissibility, 75.5% (95% CI: 73.0% - 77.6%) in attack rate, 41.5% (95% CI: 13.9% - 55.7%) in peak magnitude, and 63.1% (95% CI: 59.3% - 66.3%) in cumulative influenza-associated hospitalizations during the winter-spring period of the 2019/2020 season in Hong Kong. The implementation of PHSMs to control COVID-19 had a substantial impact on influenza transmission and associated burden in Hong Kong. Incorporating information on factors influencing influenza transmission improved the accuracy of our predictions.

Austria). The data are publicly available at https://www.chp.gov.hk/en/static/24015.html and https://www.chp.gov.hk/en/statistics/data/10/641/642/2274.html; Codes are available at the author's Github site: https://github.com/DanielYCLau/HKFluForcast2020/.

**Funding:** This project was supported by the Health and Medical Research Fund (project no. 18171202, S.T.A.); a commissioned grant from the Health and Medical Research Fund from the Government of the Hong Kong Special Administrative Region; the Research Grants Council of the Hong Kong Special Administrative Region, China (project No. T11-712/19N, B.J.C.); AIR@InnoHK administered by Innovation and Technology Commission; Shenzhen-Hong Kong-Macau Science and Technology Project (Category C) (Project no: SGDX20230821091559022, Z.D.). The funding bodies had no role in study design, data collection and analysis, preparation of the manuscript, or the decision to publish.

**Competing interests:** I have read the journal's policy and the authors of this manuscript have the following competing interests: BJC received honoraria from AstraZeneca, Fosun Pharma, GSK, Haleon, Moderna, Novavax, Pfizer, Roche, and Sanofi. The authors report no other potential conflicts of interest.

## Author summary

In theory, better forecasts or projection of influenza depends on understanding the impacts of these drivers and interventions, and their incorporation could empower the predictive performance of the underline models. Hong Kong has intensive surveillance system for influenza and a sub-tropical settings to provide more representative estimates of the impact of PHSMs on influenza. In this study, we developed a statistical model to not only forecast influenza activity and hospitalizations while considering potential associations with transmission drivers but also project influenza activity and hospitalizations retrospectively under a counterfactual scenario without COVID-19 PHSMs since January 2020. This allowed us to assess the impact of PHSMs on influenza during the winter-spring period of the 2019/20 season in Hong Kong estimating the reduction in transmissibility by 45%, attack rate by 76%, peak magnitude by 42%, and influenza-associated hospitalization burden by 63%. This is the first study to use the forecast model to assess the impact of PHSMs on influenza by exploring retrospective forecasts of different diseases outcomes even in sub-tropical setting.

## Introduction

Seasonal influenza virus causes significant morbidity and mortality, accounting for 3 to 5 million severe infections and 290,000 to 650,000 deaths each year circulating in almost every location and country across the globe [1,2]. While influenza epidemics typically occur in temperate regions during the winter season, they can occur at any time of the year with summer and winter peaks in subtropical and tropical regions [3], making it challenging for robust and accurate prediction and forecast of upcoming epidemics. Several intrinsic and extrinsic factors might have contributed to the underlying mechanism of such irregular and year-round influenza epidemics in subtropical locations including Hong Kong [3–5].

Intrinsic factors such as genetic mutations, viral antigenic drift and shift, host immune response [6–8], and interactions among co-circulating influenza viruses (types and subtypes) allowing partial immunity to same type/subtype [9–15] play a crucial role in determining influenza virus transmission and seasonality. In addition, the scheduled seasonal influenza vaccination programs can redefine population immunity for influenza seasons [16] and shape influenza phylodynamics and epidemic dynamics [17–19]. Extrinsic factors, such as climatic factors, also have a significant impact on influenza transmissibility and contribute to the seasonality of influenza epidemics. For example, ambient temperature and absolute humidity directly affect virus survival and therefore influenza transmissibility and seasonality [3,20,21]. In Hong Kong, absolute humidity (U-shape form of association with transmissibility), ambient ozone concentration (nonlinear negative association with transmissibility), and public health social measures (PHSMs) like school closures are associated with influenza transmission and can predict the bi-annual peaks of seasonal epidemics over the years [22,23]. In addition, it is possible that larger oscillations in incidence may be caused by small seasonal changes in the influenza transmission rate (by these extrinsic factors) that are amplified by dynamical resonance [24].

Understanding these factors' mechanisms and their impact on influenza transmission dynamics can provide better prediction and real-time forecasting, which can be public health tools to mitigate the burden of influenza. However, during the COVID-19 pandemic, there was minimal influenza activity around the world, which was thought to be associated with the impact of PHSMs in mitigating the COVID-19 pandemic [23,25–28]. In this study, we used

syndromic and virological multi-stream data to construct a statistical regression-based modeling framework for retrospectively forecast winter influenza activity and associated hospital admission burdens in Hong Kong during the COVID-19 pandemic first, and then evaluated the impact of COVID-19 PHSMs on influenza activity in Hong Kong.

## Results

### Influenza activity and drivers in Hong Kong

In Hong Kong, influenza viruses exhibited year-round activity and caused epidemics during the winter-spring period in most seasons and irregular outbreaks with lower magnitude in summer (Fig 1A). However, during the 2019/20 winter season, the influenza outbreak and associated hospital admissions burden significantly suppressed to a very low level from February onwards, with almost no activity recorded since March 2020 (Fig 1A and 1B). The mean temperature and absolute humidity in Hong Kong showed strong seasonality, with the highest values in summer and the lowest in winter (Fig 1C and 1D). The ambient ozone concentration in Hong Kong fluctuated over time, with lower levels during both summer and winter (Fig 1E). Besides, the summer holidays mostly aligned with the summer peaks of influenza activity and hospitalizations in Hong Kong (Fig 1A and 1B).

### Performance on short-term prediction and forecast of influenza activity

We formulated generalized linear regression models considering various potential driving factors as predictors with their possible forms of associations to forecast the weekly influenza activity in Hong Kong, assuming the weekly ILI+ proxy followed a negative binomial distribution (see MATERIALS AND METHODS section for details). The models with the best short-term prediction performance (1–4 weeks ahead) in the period 2017–2019 were shown in Table 1. The model with the lowest WIS (WIS-based model) considered the ILI+ proxy from the previous 23 weeks, monthly spline, ozone concentration and school holiday as predictors. Specifically, low ozone concentration and longer school holidays/closure were associated with high ILI+ proxy in the short term (Fig A in S1 Text). The measure goodness-of-fit ($R^2$) was 0.90 for the prediction 1 week ahead and gradually dropped to around 0.50 for the prediction 4 weeks ahead (Fig B in S1 Text). Additionally, the models that included temperature additionally found the lowest root mean square error (RMSE), root mean square log error (RMSLE) and mean absolute error (MAE) values (Table 1).

Assuming the influenza activity in January 2020 was not influenced by PHSMs prior to the introduction of index case in Hong Kong, the maximum incidence rates could range from 1.0% (95% Prediction Interval (PI): 0.2% - 2.3%) to 1.3% (95% PI: 0.3% - 3.0%) based on the short-term forecast updated by week since 1st– 4th week of January 2020 (Table 2) based on WIS-based model. Besides, the short-term forecasted incidence rates in RMSE-/RMSLE-/MAE-based model were slightly lower in magnitude compared to that in WIS-based model (Table A and Fig C in S1 Text).

### Performance on medium-term prediction and forecast of influenza activity

For the medium-term prediction of 1–13 weeks ahead (Table 1), the WIS-based model demonstrated better performance with the lagged ILI+ proxy up to 15 weeks, monthly spline, mean temperature and school holiday/closure as the predictors. Comparatively, the RMSLE- and MAE-based models also included absolute humidity as predictors, while the RMSE-based model shared a similar model as WIS-based model but with longer lags (up to 17 weeks) for ILI+ proxy. The mean temperature had a negative association with ILI+ proxy for the

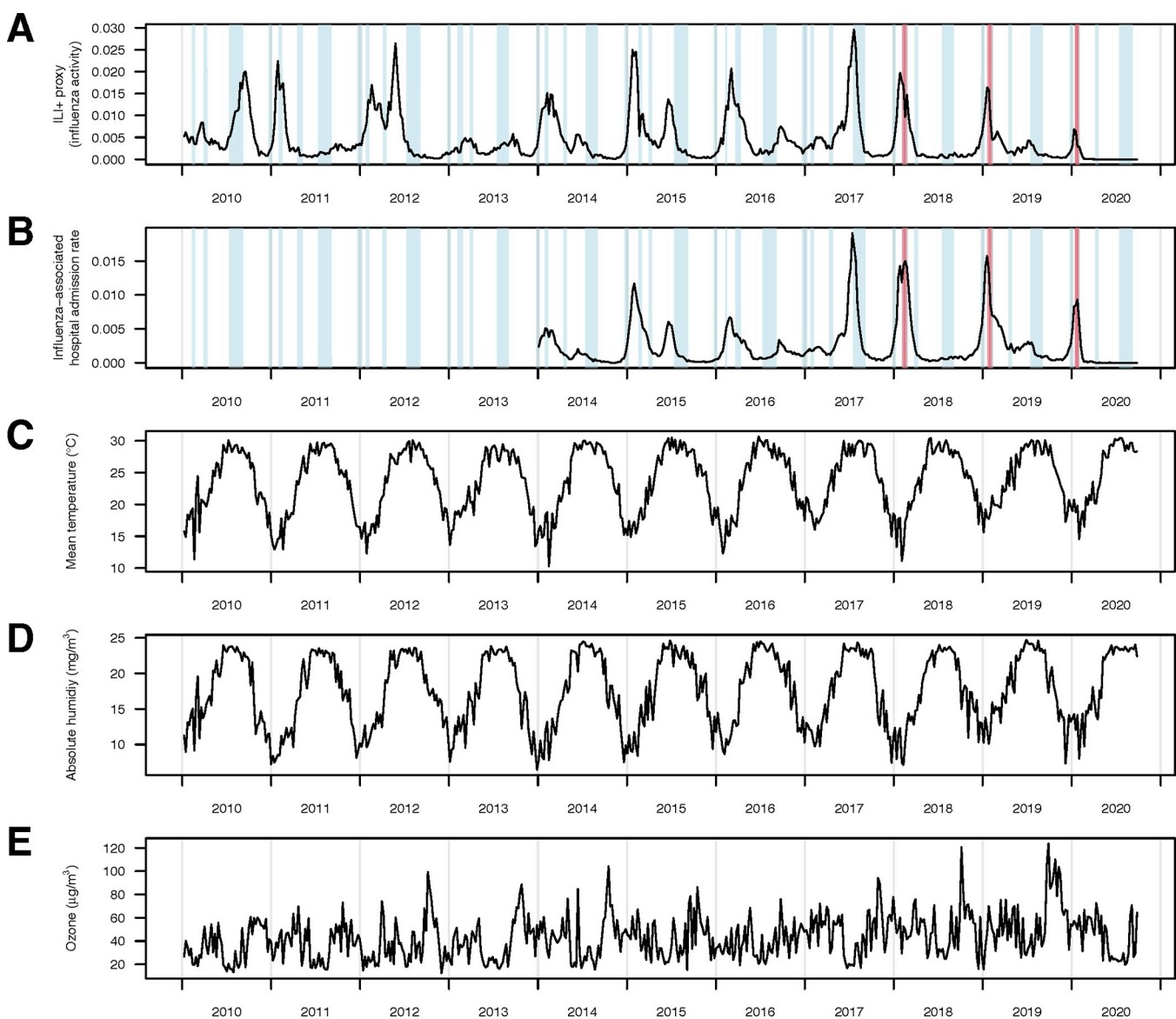

**Fig 1.** (A) Weekly influenza activity (as ILI + proxy) are in black lines from January 2010 through September 2020 in Hong Kong. The holiday-related and reactive school closures are presented in light blue bars and light red bars respectively. (B) Weekly admission rates in public hospitals with principal diagnosis of influenza from January 2014 through September 2020 in Hong Kong. (C to E) The weekly time series of mean temperate, mean ambient ozone concentration and mean absolute humidity in Hong Kong during the study period respectively. The absolute humidity was evaluated using the weekly time series of mean temperature and relative humidity.

following week, while long school holiday/closure were associated with high ILI+ proxy (Fig D in S1 Text). Besides, the $R^2$ for the prediction 1–4 weeks ahead decreased from 0.89 to 0.46, whereas $R^2$ ranged 0.24–0.39 for the prediction of further weeks ahead up to 13 weeks (Fig B in S1 Text).

Similarly, the medium-term forecast (starting from 3rd week of January 2020) suggested an influenza epidemic in the winter-spring period, with a peak in the first week of February (Fig 2B). The winter-spring epidemic would peak with the estimated weekly incidence rate of 1.2% (95% PI: 0.2% - 2.8%), resulting in a 41.5% (95% Confidence Interval (CI): 13.9%– 55.7%) reduction in the mean peak magnitude. The estimated attack rate would be 11.9% (95% PI:

**Table 1. Prediction performance of models on influenza activity based on time series cross-validation.** The models with the lowest cross-validated weighted interval score (WIS), root mean square error (RMSE), root mean square log error (RMSLE), and mean absolute error (MAE) were shown. The model with the lowest WIS was selected for further forecasting (bolded). The superscript 2 indicates the inclusion of both quadratic and linear terms of the corresponding covariate in the model. ILI(n): lagged ILI+ proxy up to n weeks; AH: absolute humidity; Temp: temperature; Ozone: Ozone concentration; School: School holiday/closure.

| Rank (cost value) | Model | WIS | RMSE | RMSLE | MAE |
|---|---|---|---|---|---|
| **Short-term (1–4 weeks)** | **ILI(23) + monthly spline + log(Ozone) + log(School)** | **1 (997.5)** | 14 (1804.8) | 1639 (0.5) | 19 (1579.7) |
| | ILI(23) + monthly spline + log(Ozone) + Temp + log(School) | 3 (998.3) | 1 (1796.3) | 1254 (0.5) | 1 (1569.6) |
| | ILI(9) + monthly spline + log(Ozone) + Temp + log(School) | 1736 (1025.8) | 1759 (1867.2) | 1 (0.4) | 1541 (1632.7) |
| **Medium-term (1–13 weeks)** | **ILI(15) + monthly spline + Temp + School$^2$** | **1 (1536.5)** | 6 (3155.6) | 1259 (0.8) | 4 (2527.2) |
| | ILI(17) + monthly spline + Temp + School$^2$ | 10 (1540.7) | 1 (3152.3) | 2559 (0.8) | 8 (2531.5) |
| | ILI(5) + monthly spline + AH + Temp + School$^2$ | 664 (1560.6) | 1103 (3224.2) | 1 (0.8) | 1188 (2585.1) |
| | ILI(15) + monthly spline + AH + Temp + School$^2$ | 14 (1541.5) | 21 (3164.8) | 120 (0.8) | 1 (2524.9) |

8.3% - 16.5%) from December 2019 to March 2020 (Table 2), compared to a median attack rate of 13.7% (range: 4.4% - 17.6%) in the winter-spring period during the seasons 2011/12-2018/19. Hence, we could infer that the COVID-19 PHSMs potentially led to an overall reduction in attack rate up to 75.5% (95% CI: 73.0% - 77.6%) during the winter-spring period in 2019/20 seasons. The medium-term forecasted results, including peak incidence, attack rate, and reduction in attack rate, were similar across different models (Table B and Fig E in S1 Text). On the other hand, the effective reproduction number dropped below 1 in early February 2020, corresponding to the observed peak in early February 2020, and achieved a maximum reduction in effective reproduction number of 44.6% (95% Credible Interval: 38.6% - 51.9%) (Table C and Fig F in S1 Text).

## Model performance for prediction and forecast of influenza-associated hospital admission rates

We also extended our method to forecast influenza-associated hospital admission rate (MATERIALS AND METHODS). The WIS-/MAE-based model demonstrated better short-term forecast performance with the lagged admission rate up to 10 weeks, monthly spline, absolute humidity, mean temperature and school holiday/closure. For medium-term forecast, a similar model with lagged admission rate up to 9 weeks was adopted (Table D in S1 Text).

**Table 2. Forecasted influenza incidence, attack rate and reduction in attack rate from cross-validated models under the counterfactual scenario without COVID-19 pandemic waves in season 2019/2020.** The forecasts were based on the model with the lowest cross-validated weighted interval score (WIS). Forecasts were updated by week since the first week of January 2020 until the first COVID case was found before the last week of January. Attack rate was defined as the cumulative sum of weekly incidence in the winter-spring period (December 2019 –March 2020), where the reduction in attack rate was defined as the percentage reduction between the observed and the forecasted attack rate. ^PI: prediction interval; *CI: confidence interval.

| Estimates, % (95% ^PI / *CI) | | Forecast Since | | | |
|---|---|---|---|---|---|
| | | **1st week of Jan 2020** | **2nd week of Jan 2020** | **3rd week of Jan 2020** | **4th week of Jan 2020** |
| **Short-term (1–4 weeks)** | max. Incidence^ | 1.0 (0.2–2.3) | 1.1 (0.2–2.6) | 1.3 (0.3–3.0) | 1.0 (0.2–2.3) |
| **Medium-term (1–13 weeks)** | Peak Time | 23rd– 29th February | 26th January– 1st February | 2nd– 8th February | 9th– 15th February |
| | Peak Incidence^ | 1.2 (0.1–3.6) | 1.1 (0.2–2.6) | 1.2 (0.2–2.8) | 0.9 (0.2–2.3) |
| | Reduction in mean peak incidence* | 40.9 (-1.5–58.3) | 36.4 (6.7–51.8) | 41.5 (13.9–55.7) | 26.9 (-3.4–43.5) |
| | Attack rate^ | 12.0 (7.8–17.4) | 10.9 (7.3–15.2) | 11.9 (8.3–16.5) | 10.3 (7.4–13.8) |
| | Reduction in attack rate* | 75.7 (72.7–78.1) | 73.2 (70.3–75.7) | 75.5 (73.0–77.6) | 71.9 (69.5–73.9) |

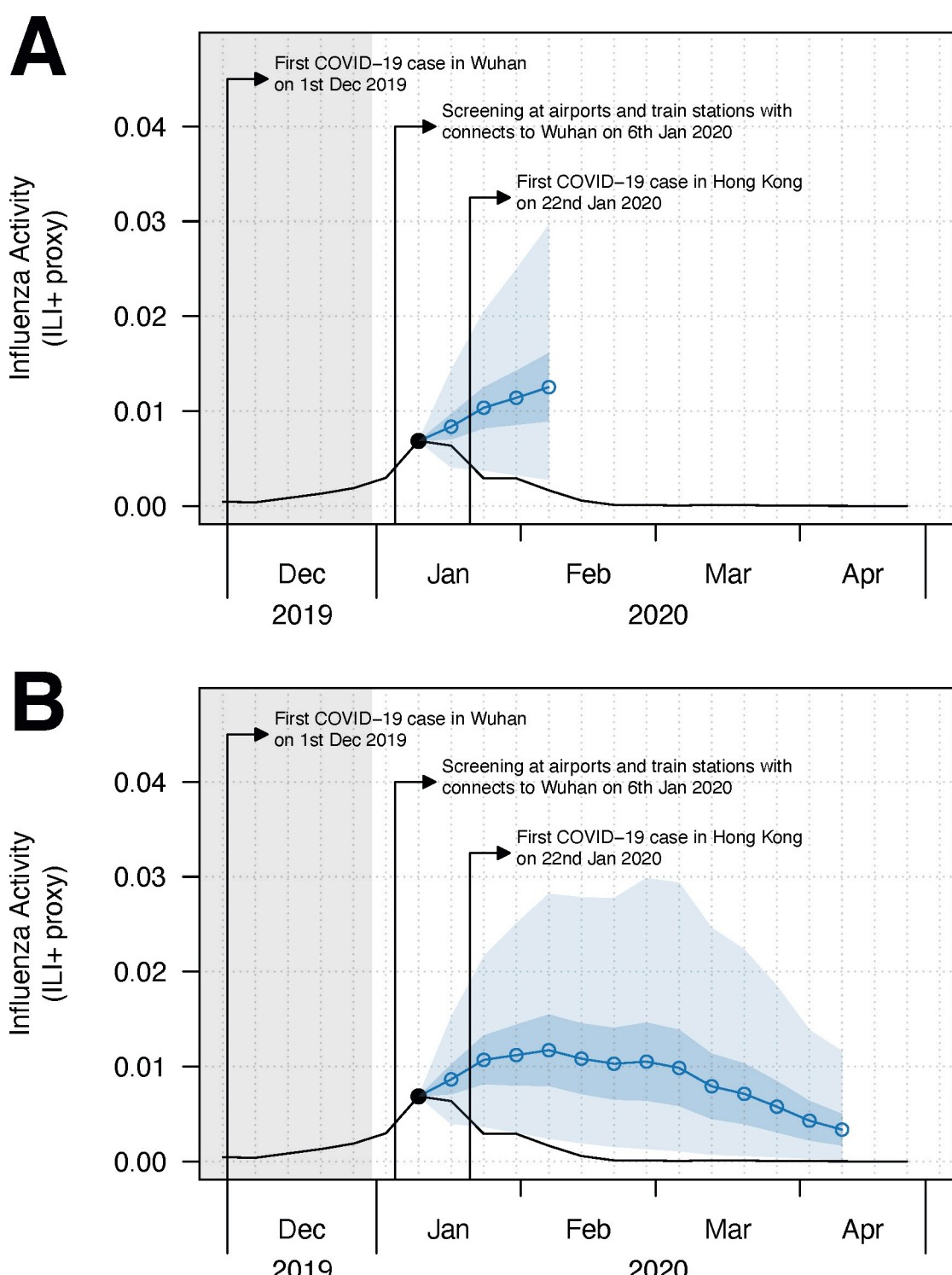

**Fig 2. (A) Short-term (1–4 weeks ahead) and (B) medium-term (1–13 weeks ahead, accounted for the whole 2019–20 winter season) forecast on influenza activity since the 3rd week of January 2020 in Hong Kong.** We considered by 2nd week of January 2020, there were changes in the population behaviour at individual and community levels and the implication of PHSMs became much effective [28], therefore we started forecasting from the following weeks (for main text). The black line is the observed ILI + proxy and the blue line (with dot shapes) is the influenza activity forecast with 95% CI (in blue shade) and 95% PI (in lighter blue shade) based on the model with the lowest cross-validated WIS. The grey area indicates the testing period of the model.

Similar to the short-term forecast for influenza activity, the forecasted hospital admission rates also showed a drop in early February 2020 (Fig 3A, Fig G and Table E in S1 Text). Besides, the medium-term forecast suggested the hospital admission rate would peak in mid-February (Fig 3B, Fig H and Table F in S1 Text). From the forecast starting from 3rd week of January 2020, the cumulative admission rate would reach 12.9% (95% PI: 9.6% - 16.8%) under the counter-factual scenario and hence led to a 63.1% (95% CI: 59.3% - 66.3%) reduction in the cumulative admission rate during the winter-spring season in 2019/20 (Table F and Fig H in S1 Text). We observed similar results in forecasting patterns and peak timing for hospital admission rates when approximating the admission rate forecast by scaling the forecasted influenza activity. However, we found that the peak magnitudes would be higher than that obtained through direct forecasting by 1.28 (95% CI: 0.76–2.08) (Fig I in S1 Text).

## Sensitivity analysis with alternative models (ARIMA models) for comparing forecast performance

We also conducted sensitivity analysis on the forecasting performance by conducting the forecast based on ARIMA models, which was another comparative statistical approach for time series modelling (see MATERIALS AND METHODS section for details). Our result suggested that GLM had better forecast performance of influenza activity than ARIMA models in general, while ARIMA model might perform slightly better in medium term in terms of point prediction (i.e. RMSLE and MAE) (Table G in S1 Text). Besides, the forecast performance of influenza-associated hospitalization rate between GLM and ARIMA model were comparable (Table H in S1 Text).

## Discussion

Unlike countries in temperate region with strong seasonal winter influenza epidemics, subtropical/tropical countries including Hong Kong have year-round influenza activity [3]. Such less regular influenza dynamics in Hong Kong are often driven by various intrinsic and extrinsic factors [22,23,29–31].

Our results suggested that ozone concentration and school holidays were crucial for the short-term forecast (1–4 weeks ahead) of influenza activity in Hong Kong, and this was consistent across all cross-validated models. Our results show a negative association between ambient ozone concentration and influenza activity (ILI+ proxy) in the upcoming weeks (Fig A in S1 Text), which could be attributed to the ozone-primed immunity under high ozone centration against influenza virus infection [22,32]. While previous studies have reported that school closure could reduce influenza transmission [23,33], our result showed a positive association between longer school closures/holidays and the ILI+ proxy in the short-term, and such association became stronger on the prediction of 4 weeks ahead compared to the prediction of 1 week ahead (Figs A and D in S1 Text). This might illustrate the possible increase in influenza activity when the school resumed classes, or due to the increased social interactions during holiday gatherings. Despite the reduction in reproduction number during weekends and school holidays, it was found in European countries that intergenerational mixing contacts were more frequent during these periods, while same-age mixing became less frequent [34]. A similar mixing pattern was also observed during the influenza A/H1N1 pandemic [35], or even during the COVID-19 pandemic despite a different context with the significantly stricter PHSMs in place [36]. This findings suggested that the decline in the incidence among children was mitigated to some extent by a corresponding rise in the incidence rate among adults. Our result reiterated the observation of summer influenza outbreaks even students were having summer holidays, and it is commonly observed in subtropical/tropical countries [3].

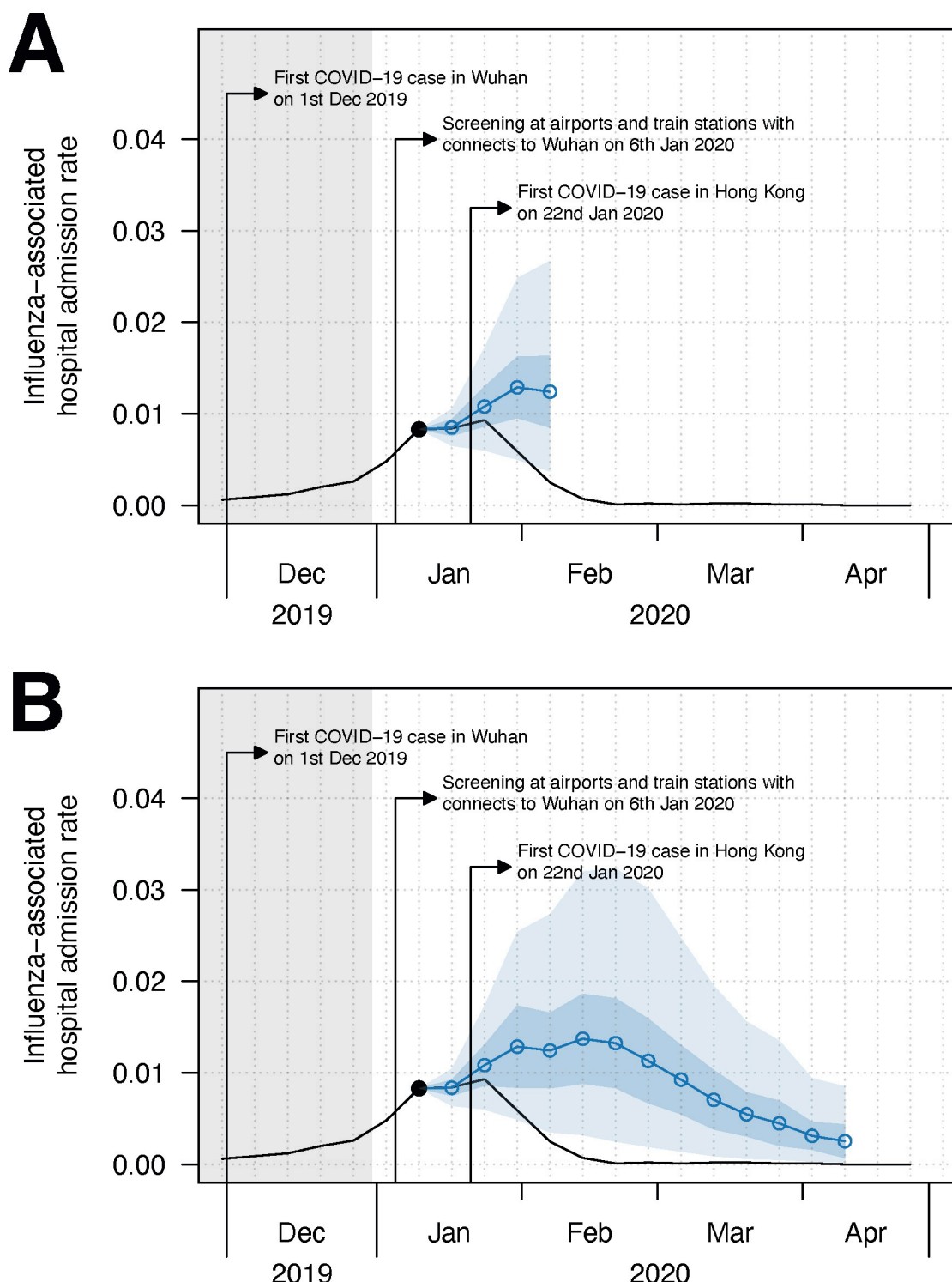

**Fig 3. (A) Short-term (1–4 weeks ahead) and (B) medium-term (1–13 weeks ahead, accounted for the whole 2019–20 winter season) forecast on influenza-associated hospital admission rate since the 3rd week of January 2020 in Hong Kong.** We considered by 2nd week of January 2020, there were changes in the population behaviour at individual and community levels and the implication of PHSMs became much effective [28], therefore we started forecasting from the following weeks (for main text). The black line is the observed ILI+ proxy and the blue line (with dot shapes) is the influenza activity forecast with 95% CI (in blue shade) and 95% PI (in lighter blue shade) based on the model with the lowest cross-validated WIS. The grey area indicates the testing period of the model.

Mean temperature not only led to an improvement in point prediction for some models in short term (Table 1), but was also a key driver for medium-term forecast (13 weeks ahead) of influenza activity in Hong Kong with negative association (Table 1 and Fig A in S1 Textt). The negative association between temperature and influenza activity might be explained by the contribution to stability of the influenza virus particles at low temperature [37], and that low temperatures might promote indoor crowding and hence favour transmission via increased host contact [38]. These factors might suggest temperature as a strong predictor of influenza seasonality especially in high latitudes [3]. Besides, absolute humidity was considered in RMSLE-/MAE-based model for medium-term forecast in positive association (Fig D in S1 Text). Despite an U-shaped association reported between absolute humidity and influenza virus transmissibility in Mainland China and Hong Kong [23] and elsewhere [39], and low absolute humidity also favoured influenza virus survival and transmission even in temperate locations [39,40], the positive association found in our study was probably confounded by mean temperature especially for Hong Kong located in sub-tropical region [39]. Indeed, absolute humidity, temperature and influenza interacted nonlinearly and further study was required to gain a more comprehensive understanding of their relationships [39].

Likewise, absolute humidity, mean temperature and school holiday/closure were potential predictors for the both short-term and medium-term forecast of influenza-associated hospital admission rate (Table D in S1 Text). It was found in South Korea that most of the respiratory patients visiting emergency departments in 2015–2017 were diagnosed with acute upper respiratory infections, influenza, and pneumonia, and they were highly associated with particulate matter ($PM_{10}$) and temperature [41]. Another study found that a high hospitalization rate was linked to the school holiday/closure in the United Kingdom, possibly due to increased contact between elderly individuals and potential influenza carriers such as children [42]. This might suggest that effect of absolute humidity, temperature and school holiday/closure on influenza activity would further extend to influenza-associated hospital admission rate and caused several disease burden in Hong Kong. These findings suggested that the impact of absolute humidity, temperature, and school holiday/closure on influenza activity might extend to influencing the hospital admission rate associated with influenza in Hong Kong. Yet, the impact of these predictors on influenza activity and hospitalization could differ, particularly when examining them from a forecasting perspective. In the case of hospitalization, shorter time lags and absolute humidity were considered to be more relevant compared to their influence on influenza activity in the medium-term forecast. This might explain the discrepancy in the forecasted hospitalization by scaling the forecasted influenza activity (with a scaling factor derived from historical information) over the direct forecast (Fig I in S1 Text).

Our study also estimated a significant reduction in influenza attack rate (75.5% (95% CI: 73.0% - 77.6%)), and transmissibility (44.6% (95% CI: 38.6% - 51.9%)) in the winter-spring period of the 2019/20 season (Table 2, Table B and C and Fig F in S1 Text), leading to a reduction in influenza-associated hospital admission burden by 63.1% (95% CI: 57.6%– 65.3%) (Table F in S1 Text). This reduction could be attributed to the strong implementation of COVID PHSMs. Following by confirmation of the index case in Hong Kong in late January 2020, the Hong Kong government implemented various public health interventions, including border restrictions, quarantine and isolation of cases and contacts, and changes in population behaviour such as social distancing and mask wearing [28]. These COVID-19 PHSMs were shown to reduce the transmissibility of influenza A H1N1 by 44% (95% CI: 34% - 53%) in early February 2020 in Hong Kong, which could explain the nearly zero ILI+ proxy recorded since March 2020 (Fig 1A). The reduction in influenza activity and associated disease burden were also observed globally during the 2019/20 winter season [43]. It was reported a reduction in influenza transmission of nearly 80% in China and 67.2% in the United States as of March

2020 associated with COVID-19 PHSMs [26], which was consistent with our result in Hong Kong during the winter-spring period (Table 2).

The potential implications of our forecasting framework could lead to early detection of the upcoming influenza outbreak in the short term. The models showed satisfactory goodness-of-fit for short-term prediction of 1–4 weeks with the $R^2$ ranging 0.50–0.90 (Fig B in S1 Text). This allowed our model to provide more accurate forecasts of influenza activity for the coming month and to inform timely disease control measure if an upcoming outbreak is detected in advance. Yet, the low $R^2$ for the prediction beyond 4 weeks limited the confidence of medium-term forecasts, unless different predictors were allowed to be responsible for the prediction of each week ahead at the expense of model interpretation. Therefore, along with the impact assessment of COVID-19 PHSMs via counterfactual retrospective forecasting, the data-driven framework could provide better real-time mitigation policy. The framework could conveniently be extended to prospective forecast the influenza activity and could be efficiently adopted for other directly transmitted diseases in Hong Kong and other locations. Besides, our framework demonstrated superior forecast performance in comparison to the ARIMA model (Table G in S1 Text), particularly in predicting influenza activity with longer training period, which could be attributed to the greater flexibility inherent in the GLM model.

There are several limitations in our study. First, the ILI+ proxy was subject to health-seeking behaviour and the capacity of laboratory surveillance, and influenza-associated hospital admission was subject to hospital admission capacity. Disease outbreaks of the influenza virus and COVID would raise public awareness and encourage heath-seeking behaviour, while the capacity of laboratory surveillance was challenged as the simultaneous surveillance for COVID-19, and the same for the associated admissions. Second, our predictive framework is based on regression models with autoregressive components and thus sensitive to the data. For instance, the presence of zero measurements in ILI+ proxies during the period with COVID-19 PHSMs might probably lead to forecasts with very low ILI+ proxies, while compartmental models allowed mechanistic simulation on the ILI+ proxy when the COVID PHSMs were relaxed. Third, this study was not stratified by the subtypes of influenza viruses, though our method can be easily applied to the subtypes of influenza viruses. Fourth, we did not considered the time-varying information on age-specific dynamics, antigenic variations, and changes in population immunity (including vaccination and cross-immunity), which could improve the predictive models and hence better forecasts of the outcomes. Finally, our result did not imply causal relationship between the predictors and influenza activity. Our statistical model could investigate the forecast performance of potential predictors on influenza activity by their correlation, whereas further studies and experimental design were required to provide stronger evidence of the causality between them.

Our study provided a data-driven statistical framework to predict and forecast less regular dynamics of seasonal influenza virus infections and hospital admissions burden in Hong Kong, which allowed early detection of the influenza outbreak and enabled timely decision-making for the public health policymakers with better healthcare preparedness including stockpiling, implementation of better interventions (PHSMs) and vaccination scheme to mitigate the upcoming epidemics. Such a forecasting framework also has the potential to quantify the direct/indirect impact of interventions for similar directly transmitted diseases (e.g. COVID-19 and influenza) on their associated burden of infections and hospitalization.

## Materials and methods

### ILI and influenza-associated hospitalization rates time series

The Centre for Health Protection in Hong Kong has established a sentinel surveillance network of approximately 50–60 private medical practitioners to monitor seasonal influenza

activity. Most acute ambulatory care in Hong Kong is delivered through the private sector. Weekly consultation rates of outpatients presenting influenza-like illness (ILI), defined as fever $\geq 38.0°C$ plus cough and/or sore throat, were reported. Additionally, laboratory testing for influenza viruses was conducted by the Public Health Laboratory Services Branch on samples submitted mainly from local hospitals and the sentinel outpatient physicians. We evaluated the weekly records of ILI consultation rate and proportion of specimens that tested positive for influenza virus from January 2010 to September 2020, excluding the 2009 influenza A(H1N1)pdm09 pandemic. Proxy measures of influenza virus activity in the community were obtained by multiplying the weekly ILI rates with the weekly proportion of influenza-positive specimens, denoted as ILI + proxy [44]. This proxy measure was previously shown as a very close correlate of laboratory-confirmed H1N1pdm09 hospitalizations in Hong Kong in 2009–10 [45]. The proxy was further scaled up by $10^6$ to approximate the incidence number of influenza infections in a million consultation [22,23,29,30,46,47]. The admission rate in public hospitals with principal diagnosis of influenza was retrieved from the sentinel surveillance system, updated weekly since 2014.

## Meteorological and pollutant data

We previously found that absolute humidity and ozone are potential extrinsic meteorological drivers for influenza transmissibility in Hong Kong [22,23,48]. We retrieved daily mean air temperature and mean relative humidity for Hong Kong during the study period from the Hong Kong Observatory [49], and daily concentration data on ambient ozone from the Hong Kong Environmental Protection Department during the study period [50]. The daily mean absolute humidity was derived from the mean relative humidity and mean temperature [51].

## School holidays and closures

Schools can be an important location for influenza transmission, particularly among school-aged children, who are one of the high-risk groups, can be more susceptible to influenza virus infection than adults [52], and tend to have more social contacts than other age groups [34,53]. Therefore, school holidays and closure are considered as one of the public health and social measures (PHSMs) to reduce the social contact and transmission between students, and hence mitigate influenza epidemics in the community [29,33]. We first reviewed the timing of school holidays for local government schools in Hong Kong, which are attended by more than 95% of school-age children in Hong Kong, and identified the dates of school holidays during Christmas and the calendar New Year (from the end of December to the start of January), the Chinese New Year holidays (from the end of January to the start of February), the Easter holidays (from the end of March to the start of April), the summer holidays (from mid-June to the end of August), as well as public holidays and weekends during the study period. We further included the timing of all reactive school closures during the epidemics in Hong Kong.

## Statistical modelling framework for forecast

We considered the ILI+ proxy to follow a negative binomial distribution and proposed a predictive framework under a series of generalized linear models (GLMs) as below,

$$\log(\lambda(t + k)) = \boldsymbol{\alpha_k}\boldsymbol{B}(t - 1) + \sum_{l=1}^{L} \beta_{k,l} y(t - l) + \sum_{j=1}^{n} \psi_{k,j}(t - 1)$$

Where $\lambda(t+k)$ is the expected ILI+ proxy for the coming $k$ week with $k \in \{0,1,2,3\}$ at week $t$. $\boldsymbol{B}(t)$ is the basis vector of week $t$ from the periodic cubic basis spline matrix with the period of

a year, and $\boldsymbol{\alpha_k}$ is the spline coefficient vector. We assumed the number of basis functions to be 12 within a period of a year which corresponded to a cubic basis spline assigned to every month (i.e. monthly spline). $y(t-l)$ is the log observed ILI+ proxy at week $t-l$ and $\beta_{k,l}$ is the associated coefficient. The observed ILI+ proxy with zero count was set as 0.5 when acting as the predictor. $\psi_{k,j}(t-1)$ is the respective form of association for $n$ potential extrinsic seasonal drivers (including climatic, environmental and socio-demographic drivers) of influenza virus transmission. Different nested forms (i.e., Linear form: $\psi_{k,j}(t) = \gamma_{k,1,j}C_j(t)$, U-shaped/ quadratic form: $\psi_{k,j}(t) = \{\gamma_{k,1,j}C_j(t) + \gamma_{k,2,j}C_j^2(t)\}$ and non-linear power form: $\psi_{k,j}(t) = \gamma_{k,1,j}\log(C_j(t))$) were explored and the best form was determined to incorporate for each potential drivers as covariates in the model. Here, $C_j(t)$ is the $j$th covariates at week $t$ and $\gamma_{k,1,j}$ and $\gamma_{k,2,j}$ are respective coefficients. We considered the splines function ($\boldsymbol{B}(t)$) constructed by the seasonal pattern of influenza activity in the previous seasons, hence we set $L = 26$ (i.e. approximately half year) for model training. We included mean temperate, absolute humidity, ozone concentration, and school holidays/closure as covariates in the model with their respective form $\psi_{k,j}(t)$ as the extrinsic drivers for influenza virus transmission. Besides, an overdispersion parameter ($\phi_k$) was introduced to control the variance for the forecast at $k$-th week, and the forecast would reduce to be Poisson-distributed if there was no overdispersion. Model parameters were estimated by the maximum likelihood method. Statistical analyses were conducted using R version 4.1.1 (R Foundation for Statistical Computing).

## Short-term prediction and forecast of influenza activity

To assess the prediction performance of the models for forecasting of influenza activity, we conducted a time series cross-validation during the period 2010–2019 for model selection. Under a 70/30 training-to-testing ratio, we used data from January 2017 to December 2019 as testing period, hence we fixed the training period for 7 years (i.e from January 2010 to December 2016 as the first training window) and evaluated the data in the coming $k$ weeks (i.e. 1st- $k$th week in 2017 as the first testing window) using a rolling window approach. We then evaluated the short-term forecasting performance (1–4 weeks ahead) using the weighted interval score (WIS). The WIS considered both the error in point predictions and prediction intervals, providing a probabilistic perspective on forecast accuracy [54,55]. We considered 11 prediction intervals with corresponding nominal coverages of 98%, 95%, 90%, ..., 10% to calculate WIS, which was the forecast evaluation used in the *COVID-19 Forecast Hub* [55]. As a sensitivity analysis, we also considered the traditional measures like root mean square error (RMSE), root mean square log error (RMSLE) and mean absolute error (MAE) which only considered point prediction. Models with lower values of WIS, RMSE, RMSLE and MAE implied better prediction performance, and the selected predictors (e.g. ILI+ proxy up to different lags, different extrinsic drivers) would vary depending on the cost function. The coefficient of determination ($R^2$) for the prediction of each week ahead in the training data would also be assessed.

The prediction (or forecasting) uncertainty was evaluated by sampling 10,000 sets of model coefficients given the asymptotic normality of the maximum likelihood estimator and were thus used to generate samples of predicted ILI+ proxy. The 2.5th and 97.5th quantiles of the predicted samples were considered as the 95% prediction interval (PI). We reported the reduction in the outcomes by comparing observed and forecasted (counterfactual) scenarios with uncertainty as 95% confidence interval (CI) to illustrate the changes in mean prediction.

In Hong Kong, the control measures and public awareness against the emerging SARS-CoV-2 virus were placed well from 2nd week of January, 2020 [28]. We hence considered the following week (3rd week of January, 2020) as the forecast starting point for the related results

presented in the main text. We also allowed different starting time points from the 1st to 4th week of January, 2020 for forecast as sensitivity analysis.

## Medium-term prediction and forecast of influenza epidemic in 2020 and impact assessment of COVID-19 pandemic

We further extended our method to forecast influenza activity during winter in 2020 by fitting our regression model to the data up to 13 weeks ahead (i.e. $k \in \{0,...,13\}$, accounting for whole 2019–20 winter season), following the similar procedure as in short-term prediction to conduct model selection and obtain the prediction interval. This allowed us to estimate the incidence number, the attack rate of influenza infection and transmissibility under a counterfactual scenario with no PHSMs. The (instantaneous) effective reproduction number, a measure of transmissibility was calculated by using the "EpiEstim" package in R [56]. Hence, the impact of PHSMs on infections, the attack rate, and effective reproduction number were evaluated by comparing these measures for observed and counterfactual forecasted scenarios for the winter season (December 2019—March 2020) [23].

## Forecast of influenza-associated hospital admission burden in 2020 and impact assessment of COVID-19 pandemic

We also extended the framework to forecast influenza-associated hospital admission rates in 2020, assuming $\lambda(t+k)$ is the expected hospital rate at the $k^{th}$ upcoming week with $k \in \{0,1,2,3,...\}$ at week $t$, following a negative binomial distribution. Since the hospitalization data was available since 2014, we used data from January 2018 to December 2019 as testing period and conducted the rolling time series cross-validation with a fixed training period for 4 years (i.e. from July 2014 to June 2017 as the first training window). We performed both the short-term (1–4 weeks ahead) and medium-term (1–13 weeks ahead) forecasting of influenza-associated admission rates since the 3rd week of January 2020. Similarly, we assessed the impact of COVID-19 PHSMs on admission rates and cumulative admission rates for the winter season.

Moreover, we used historical information on the influenza-associated admission and ILI + proxy data from 2014 to 2019 to forecast the hospital admission burden in 2020 by rescaling the forecasted ILI+ proxy with calendar-week-specific means of proportions of admission rate. We hence compared the forecasting performance for hospital admission burden using these two approaches.

## Sensitivity analysis with Autoregressive integrated moving average (ARIMA) models based forecasting

ARIMA models were another popular statistical approach to conduct time series forecast. In this sensitivity analysis, we also obtained the forecasting performance given the same set of training/testing data in short-/medium-term forecast for ILI+ proxy and hospitalization admission rate. To ensure computational efficiency, we selected the predictors from the GLM that exhibited the best forecasting performance based on WIS, RMSE, RMLSE and MAE. These selected predictors were then included in the ARIMA models. Besides, to identify the appropriate AR and MA components in ARIMA models, we used the "auto.arima" function in "forecast" package in R [57] and repeated for each training window, resulting in the selection of the five most popular ARIMA models used throughout the entire training period. We thus re-trained only these five ARIMA models and evaluated their forecast performance, comparing it to that of the GLMs.

## Supporting information

**S1 Text. Supplementary Tables and Figures.**
(PDF)

## Acknowledgments

The authors thank Julie Au for technical assistance.

## Author Contributions

**Conceptualization:** Benjamin J. Cowling, Sheikh Taslim Ali.

**Data curation:** Yiu-Chung Lau, Songwei Shan, Sheikh Taslim Ali.

**Formal analysis:** Yiu-Chung Lau.

**Funding acquisition:** Zhanwei Du, Benjamin J. Cowling, Sheikh Taslim Ali.

**Investigation:** Yiu-Chung Lau, Sheikh Taslim Ali.

**Methodology:** Yiu-Chung Lau, Sheikh Taslim Ali.

**Software:** Yiu-Chung Lau.

**Supervision:** Benjamin J. Cowling, Sheikh Taslim Ali.

**Writing – original draft:** Yiu-Chung Lau, Sheikh Taslim Ali.

**Writing – review & editing:** Yiu-Chung Lau, Songwei Shan, Dong Wang, Dongxuan Chen, Zhanwei Du, Eric H. Y. Lau, Daihai He, Linwei Tian, Peng Wu, Benjamin J. Cowling, Sheikh Taslim Ali.

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
