## [Decision Letter · Decision Letter 0]

27 Mar 2024

Dear Prof Cowling,

Thank you very much for submitting your manuscript "Forecasting of influenza activity and associated hospital admission burden and estimating the impact of COVID-19 pandemic on 2019/20 winter season in Hong Kong" for consideration at PLOS Computational Biology.

As with all papers reviewed by the journal, your manuscript was reviewed by members of the editorial board and by several independent reviewers. In light of the reviews (below this email), we would like to invite the resubmission of a significantly-revised version that takes into account the reviewers' comments.

We cannot make any decision about publication until we have seen the revised manuscript and your response to the reviewers' comments. Your revised manuscript is also likely to be sent to reviewers for further evaluation.

Sincerely,

Benjamin Althouse

Academic Editor

PLOS Computational Biology

Rob De Boer

Section Editor

PLOS Computational Biology

Reviewer's Responses to Questions

**Comments to the Authors:**

Reviewer #1: Overall

*******

This is a nice study question and good evidence for the ability of the framework to forecast. It also has implications for the potential effectiveness of social distancing against influenza transmission. It builds on a sizeable literature for forecasting.

What were the models used for the different sections? The results-last organisation of the manuscript didn’t really work, because I couldn’t understand what the models were doing. I did jump ahead to the more technical methods section, but that still wasn’t fully clear. I think the regression framework was drawing in historical averages from prior seasons and adjusting those for more recent patterns for the short term forecasts, but allowing the model to regress back to historical averages for medium-term predictions. This is a nice framework and appropriate, but it is difficult for the reader to appreciate this in the current draft. I’d suggest adding in some natural language at the end of the introduction and keeping the current structure, or making the methods more accessible and less technical and switching to methods-first.

The paper suggests that reductions in social mixing less than those used in Hong Kong for COVID would be sufficient to control influenza, if that was their purpose. This seems to be an important additional point that could be made more explicitly in the paper.

No curated data are available nor is the code available. Code on request is not sufficient.

Detailed comments

Line 26: make it clear if / whether this was a retrospective study

89: understanding these factors should help. The authors cite many studies, but often including these factors doesn’t make a massive improvement.

109: “usually coincide” suggest a proof for grammar

121: either here or in the introduction, it would be good to discuss measures of forecast accuracy. Different potential customers require different outputs. Here, the forecast is being used as a scientific tool to some degree, but a bit of discussion is warranted as to why we care… perhaps calling out the categorical flood-like levels that are often used by public health professionals. Also worth having a visual comparison of accurate and inaccurate forecasts so that the reader can judge more easily the meaning of the different statistics presented in the table.

162:I find the R^2 a difficult measure with which to judge forecast accuracy. Can the authors comment a little on this so that the reader knows how to interpret it. Or change the narrative to focus on one of the measures that are more easily comparable across different forecasting studies. RMSRE at 2 weeks is a pretty good one?

180: this is a really nice exposition of a principled counterfactual. The authors should comment a little more on the need for those for historical events when setting policy options for future scenarios.

247: Is this some sense of overshoot? So if the infection becomes well seeded but can’t grown exponentially during school closure it then accelerates more and overshoots more than it would have done had schools been open when it was initially growing? There might eb a nice effect to try to tease out here a little more in the discussion if the author can dig out a few references on overshoot?

277: this implies a maximum R value for flu during that period, which is probably worth reporting

310: are there any other studies of forecasting healthcare demand in HK?

362: This section needs some natural language added to make it more accessible. Perhaps at the end of the intro or woven in here. Broadly, what signals can the framework pickup? How are they combined and is there a history of this type of framework being good for prediction?

Reviewer #2: General: This is a well-written manuscript, which investigates the ability to forecast influenza epidemics in a subtropical region via a traditional generalized linear modeling approach. However, I have a couple of concerns, which should be addressed before publication.

Major:

1. The authors should better highlight the novelty of their work. It does neither become clear what is new here from an application point of view nor from a methodological perspective.

2. The authors propose a GLM based modeling approach, assuming ILI+ to follow a negative binomial distribution. No justification is given for this choice. Furthermore, there is no comparative modeling approach such as ARIMA tested.

3. Time series cross-validation is non-trivial. Based on the description on pages 19/20 it remains unclear, how exactly training and test sets were constructed. Was a sliding training window used? If yes, which size did it have and which stride? Was an expanding window applied? Was their a gap between training and test set?

4. The authors evaluated the statistical influence of public health measures on model predictions. However, this does not tell us anything about the causality of those measures. This point should be emphasized more.

**Have the authors made all data and (if applicable) computational code underlying the findings in their manuscript fully available?**

Reviewer #1: **No: **The curated data and code should be made available through a standard source, e.g. github.

Reviewer #2: Yes

PLOS authors have the option to publish the peer review history of their article (what does this mean?). If published, this will include your full peer review and any attached files.

Reviewer #1: No

Reviewer #2: No
---

## [Decision Letter · Decision Letter 1]

10 Jul 2024

Dear Prof Cowling,

We are pleased to inform you that your manuscript 'Forecasting of influenza activity and associated hospital admission burden and estimating the impact of COVID-19 pandemic on 2019/20 winter season in Hong Kong' has been provisionally accepted for publication in PLOS Computational Biology.

Best regards,

Benjamin Althouse

Academic Editor

PLOS Computational Biology

Rob De Boer

Section Editor

PLOS Computational Biology

Reviewer's Responses to Questions

**Comments to the Authors:**

Reviewer #1: Thanks for making good use of the comments. This is a nice piece of work. I would encourage you to add in more natural language to future similar manuscripts to help readers understand what the methods are doing.

Reviewer #2: The authors have appropriately addressed my concerns.

**Have the authors made all data and (if applicable) computational code underlying the findings in their manuscript fully available?**

Reviewer #1: **No: **Not clear if the curated data are being provided. If not, the authors need to explain why.

Reviewer #2: None

PLOS authors have the option to publish the peer review history of their article (what does this mean?). If published, this will include your full peer review and any attached files.

Reviewer #1: No

Reviewer #2: No

---

## [Editor Report · Acceptance letter]

26 Jul 2024

PCOMPBIOL-D-23-02063R1 

Forecasting of influenza activity and associated hospital admission burden and estimating the impact of COVID-19 pandemic on 2019/20 winter season in Hong Kong

Dear Dr Cowling,

I am pleased to inform you that your manuscript has been formally accepted for publication in PLOS Computational Biology. Your manuscript is now with our production department and you will be notified of the publication date in due course.

With kind regards,

Dorothy Lannert
